# Health care expenditures among long-term survivors of pediatric solid tumors: Results from the French Childhood Cancer Survivor Study (FCCSS) and the French network of cancer registries (FRANCIM)

**Daniel Bejarano-Quisoboni**[1,2,3,4], **Nathalie Pelletier-Fleury**[2,3], **Rodrigue S. Allodji**[1,3,4], **Brigitte Lacour**[5,6], **Pascale GrosClaude**[7], **FRANCIM Group**[¶], **Hélène Pacquement**[8], **François Doz**[8,9], **Delphine Berchery**[10], **Claire Pluchart**[11], **Piere-Yves Bondiau**[12], **Julie Nys**[1,4], **Angela Jackson**[1,3,4], **Charlotte Demoor-Goldschmidt**[1,13], **Agnès Dumas**[14], **Cécile Thomas-Teinturier**[1,15], **Giao Vu-Bezin**[1,4], **Dominique Valteau-Couanet**[16], **Nadia Haddy**[1,3,4], **Brice Fresneau**[1,3,16], **Florent de Vathaire**[1,3,4]*

**1** Radiation Epidemiology Team, CESP, Inserm U1018, Villejuif, France, **2** Primary care and Prevention Team, CESP, Inserm U1018, Villejuif, France, **3** Université Paris-Saclay, UVSQ, Inserm, CESP, Villejuif, France, **4** Department of Research, Gustave Roussy, Villejuif, France, **5** EPICEA, CRESS, INSERM UMR 1153, Université de Paris, Paris, France, **6** Registre National des Tumeurs Solides de l'Enfant, CHRU Nancy, Vandoeuvre-lès-Nancy, France, **7** INSERM U1027, Toulouse, France, **8** SIREDO Center (Care, Research, Innovation in Pediatric, Adolescents and Young Adults Oncology), Institut Curie, Paris, France, **9** University of Paris, Paris, France, **10** Epidemiology Unit, Claudius Regaud Institute, Toulouse, France, **11** Pediatric Oncology, Reims, France, **12** Radiation Therapy, Antoine Lacassagne Cancer Center, Nice, France, **13** CHU Angers, Angers, France, **14** Université de Paris, ECEVE, UMR1123, Inserm, Paris, France, **15** Service d'Endocrinologie et Diabétologie Pédiatrique AP-HP, Université Paris Saclay, Gif-sur-Yvette, France, **16** Department of Children and Adolescent Oncology, Gustave Roussy, Villejuif, Paris, France

¶ Membership of FRANCIM-Group is provided in the Acknowledgments.
* florent.devathaire@gustaveroussy.fr

## Abstract

### Background

Childhood cancer survivors (CCS) may require lifelong medical care due to late effects of cancer treatments. Little is known about of their healthcare utilization and expenditures at long-term especially in publicly funded health care system. We aim to estimate and describe the health care expenditures among long-term CCS in France.

### Methods

A total of 5319 five-year solid CCS diagnosed before the age of 21 between 1945 and 2000 in France were identified in the French Childhood Cancer Survivors Study cohort (FCCSS) and the French cancer registry. Information about health care expenditure was taken from the French national health data system between 2011 and 2016, and was described according to survivors' characteristics. Generalized linear models were used to determine associations between health care expenditures and survivors' characteristics.

**Data Availability Statement:** The study data (FCCSS Cohort Data, which contain potentially identifying or sensitive patient information) are data that can be accessed upon request. However, there is no non-author to whom requests for access to the data can be made. The data were obtained and are managed by our team and all contact details are on the cohort website (https://fccss.fr/). On the other hand, the committee of the French National Institute of Health and Medical Research (French Acronym: INSERM), which approved the study, is a large national public institution, the member of which having no specific link with the cohort and no authorization for taking decision about data sharing. Therefore, is best to leave our team contact, for the data requests that appear on the FCCSS website: Inserm, CESP, Team Cancer and Radiation Gustave Roussy B2M 114 rue Edouard Vaillant 94805 Villejuif CEDEX Phone: 0800 804 024 (toll free) Email: contact. fccss@gustaveroussy.fr.

**Funding:** DBQ received a doctoral grant from the Paris-Saclay University (N˚ contract 9R_2019_PSU000101141_Upsud). This work was funded by the Ligue Nationale Contre le Cancer (Grant N˚RAB20035LLA). The FCCSS cohort was funded by the French Society of Cancer in Children and adolescents (SFCE), the Gustave Roussy Fondation - PSI Interval, the Foundation ARC (POPHarC program) and The French National Research Agency (ANR, HOPE-EPI project). The funders had no role in study design, data collection and analysis, decision to publish, or preparation of the manuscript.

**Competing interests:** The authors have declared that no competing interests exist.

## Results

Mean annual amount of healthcare expenditures was € 4,255. Expenditures on hospitalizations and pharmacy represents 60% of total expenditures. Mean annual of healthcare expenditures were higher at increasing age, among women survivors (€ 4,795 vs € 3,814 in men) and in central nervous system (CNS) tumor survivors (€ 7,116 vs € 3,366 in lymphoma and € 3,363 in other solid tumor survivors).

## Conclusions

Childhood cancer survivorship is associated with a substantial economic burden in France. We found that female gender and CNS primary cancer were associated with increased healthcare expenditures.

## Introduction

Childhood cancer survivorship has improved considerably in the last few decades. Nowadays, more than 80% of pediatric cancer patients reach 5-year survival, leading to a growing number of long term survivors. In Europe, it is estimated that there are between 300,000 and 500,000 former pediatric cancer patients [1].

Nevertheless, most of this progress has been obtained using treatments that can damage healthy tissues. Consequently, childhood cancer survivors (CCS) carry a significant risk of cancer treatments related late effects [2]. Main late effects are secondary neoplasms, cardiac-vascular diseases, growth problems, mental health issues, infertility and organ dysfunction [3]. Therefore, long-term after-cancer care involves lifelong medical visits and tests to prevent and manage these potential life-threatening or disabling late effects [4]. This higher burden of morbidity and surveillance of late effects is likely associated with higher costs to the health care system [5]. Moreover, previous studies have shown that some CCS could have difficulties into educational attainment and are more likely to be unable to work or to miss work days due to health conditions [6–8]. Under those circumstances, a better understanding of the healthcare use and expenditure of CCS is important to evaluate long-term consequences of survivorship.

In France, there are about 50,000 adults who had cancer in their childhood [9]. Healthcare system in France is mainly publicly funded and provides universal coverage to its citizens covering most medical expenses [10] especially for long-term conditions, however, no studies have estimated the global healthcare expenditures among CCS. The French Childhood Cancer Survivor Study (FCCSS) which includes CCS treated before 2001 provides an opportunity to detail the overall healthcare expenditures among very long-term cancer survivors. Nevertheless, FCCSS survivors were treated during their childhood in Centers for the fight against cancer (CLCC) which are specialized hospitals in cancer treatment in France. Therefore, we hypothesize that the FCCSS could include patients with more advanced and/or aggressive cancers, or they may have received more innovative treatments and consequently differ in terms of long-term outcomes and future health expenses from other CCS treated in other hospital settings.

The aims of this paper was to quantify and describe the health care expenditures among very long-term CCS in France. Subsequently, we compared the level of expenditures between FCCSS survivors with those from other settings included in the French cancer registries existing during FCCSS recruitment period.

## Material & methods

### Study population

We used data from two sources, a multicenter cohort study (FCCSS) and the French Network of cancer registries (FRANCIM).

The FCCSS is a retrospective cohort of 7,670 five-year CCS diagnosed for solid cancer or lymphoma (all malignancies except leukemia), before the age of 21 years between 1945 and 2000. Detailed information on the methods for data collection and validation has been already described [11, 12].

FRANCIM includes all the population-based registries of cancer in France. This network records all newly diagnosed and confirmed cancer cases since 1975 in diverse areas of France [13, 14]. The population covered by the FRANCIM's database represents 22% of the French population [15]. Therefore, the FCCSS survivors who were included also in FRANCIM were considered as FCCSS patients. Leukemia survivors from FRANCIM were excluded for better comparability with the FCCSS which did not include leukemia.

From both data bases, we selected all five-year solid CCS diagnosed before January 2001 who were alive in January 2011 and who were linked to the National Health Data System *(Système National des Données de Santé)* (SNDS). Since no social security number is collected in the study, patients are identified within the SNDS by probabilistic matching with the full involvement of different French healthcare-related organizations: Caisse nationale de l'assurance vieillesse des travailleurs salaries (CNAVTS) is the third party for social security number reconstruction and Caisse Nationale d'Assurance Maladie des Travailleurs Salaries (CNAMTS) is the trust third party for SNDS health data gathering based on a non-identifiable number derived from the retrieved social security number. The highly specific identification data that are provided to CNAMTS by the National Institute of Health and Medical Research (INSERM) are listed as below: family and first name, sex, date and place of birth and, unique arbitrary number. CNAMTS was in charge of both communication with CNAVTS and SNDS data extraction. The percentage of survivors linked to SNDS data after this procedure was 55.6% (n = 3786) for FCCSS and 71.9% (n = 2031) for FRANCIM survivors. In the present study, French national security number is not held by INSERM at any time.

Survivors were followed throughout the SNDS until December 2016 or death. We excluded survivors who lived outside metropolitan France during follow-up due to the difference in the health insurance system, characteristics of the population and funding of care in French overseas territories [16].

The study was approved by the French Data Protection Authority (CNIL) (Authorization n˚902287) and by the ethics committee of the INSERM. Patient informed consent was not required for this study because we obtained a specific act in law from the French "Conseil d'Etat", the highest court in France (Order 2014–96 of 2014 February 3), that approved the cession of the SNDS data for all patients included in the FCCSS and FRANCIM.

### Data sources

The SNDS is the health care claims dataset in France, which contains exhaustive individual data used for the billing and reimbursement data of the beneficiaries of the various national health insurance schemes which now covers more than 95% of the French population [17, 18].

The SNDS is mainly composed by the outpatient healthcare consumption database (*Données de Consommation Inter-Régimes database*, DCIR) and the private and public hospital database (*Programme de Médicalisation des Systèmes d'Information*, PMSI) which is divided in four categories: Medicine, surgery and obstetrics hospitalizations (MCO), home

hospitalizations (HAD), after-care and rehabilitation (SSR) and psychiatry (PSY). Although MCO and HAD had cost information since 2006, the availability of billing records for PSY and SSR systems began in 2011, thus we established this year as starting date when data was available for all systems.

Information in the SNDS includes some demographic characteristics (age, gender, place of residence); diagnosis of long-term conditions (*affections de longue durée*) defined as a disease in which the severity and/or the chronicity require a long-term costly treatment; as well as dates, nature and reimbursement of outpatient visits, dispensed medication, allied health professional visits, lab tests, medical devices, medical transports, paid sick leave and hospital admissions, including procedures performed within diagnosis-related groups [18].

## Primary measures

This study was carried out from the *Assurance Maladie* (AM) [French national health insurance] perspective (payer). Expenditures were estimated considering all the reimbursements made by the AM between January 2011 and December 2016 or date of death. Therefore, reimbursements from private health insurance, especial schemes or the final out-of-pocket were not included.

Our primary outcome variables were the total sums of direct healthcare expenditure for every calendar year for each patient in both cohorts. Direct expenditure was also classified for every year into fourteen categories: General practitioner visits, other specialist visits, physiotherapy, nursing visits, other health professionals visits, pharmacy, medical device, laboratory test, technical medical procedures, transport, hospitalizations, disability benefits, sick leave and others. All expenditures were expressed in real terms using the consumer price index with a 2015 base year provided by the National Institute of Statistics and Economic Studies (INSEE).

## Covariates

Other covariates included age, sex, year of diagnosis, age at diagnosis, type of primary cancer (kidney tumor, neuroblastoma, lymphoma, soft tissue sarcoma, bone sarcoma, central nervous system tumor, gonadal tumor, thyroid tumor, retinoblastoma and others), French deprivation index in 2009 which is an area-based multidimensional index that measures socioeconomic differences [19] and where higher scores implies a higher "deprivation" (categorized into quintiles), and death (alive or death at December 2016).

## Statistical analysis

Patient's characteristics were described along with the estimation of annual mean expenditure over 2011–2016 period. Categorical variables were expressed as numbers and percentages, and continuous variables as mean ± SD. Direct total expenditures were described by categories and also according to primary cancer.

Given the population studied, the frequency of persons having no healthcare expenses during the 6 years follow-up period was less than 4% of the total patients within each cohort group, therefore a 2-stage model was not required. Instead, we used a repeated measures generalized linear model (GLM) with a gamma distribution and a log link to estimate per-person annual medical expenditures for all patients accounting the skewness in the distribution. We added 1€ to all expenditures to allow inclusion of individuals with no expenses [20] and used age at diagnosis as categorical variable in the models due to the strong correlation with age at follow-up and year of diagnosis. Neuroblastoma was chosen as the referent for type of primary cancer variable since was one of the larger group of cancer of same histology. We compared

the output of the model adjusted by all covariates (Model I), with models excluding variable of death (Model II) and patients who died (Model III).

Finally, we performed several analysis by considering separately each type of expenditure and each type of cancer along with different interactions between cohort (FCCSS or French cancer registries) and type of cancer. Statistical significance was determined using p<0.05. All analyses were performed using SAS 9.4 software (SAS Institute, Cary, NC, USA).

## Results

### Survivors' characteristics and total health care expenditure

A total of 5,319 CCS were included in the study, among which 67.5% belong to the FCCSS (Table 1). Almost half of the patients were women and were diagnosed after 1990 (44.9% and 46.6% respectively). More than 50% of the patients were over 30 years old at the beginning of follow-up. The most common primary cancer was lymphoma (20.1%) followed by central nervous system (CNS) tumor (14.2%). Between 2011 and 2016, around 3% of the total patients had died. Details of patient characteristics by cohort are shown in the S1 and S2 Tables.

Total direct healthcare expenditure for the 5,319 patients between 2011 and 2016 were € 134,523,643. Annual mean of health care expenditures of survivors was € 4,255. However, this variable was positive skewed due to a few very high values among survivors. In detail, 50% of patients had an annual mean of healthcare expenditures lower than € 1,000 and almost 10% was above € 10,000 (S1 Fig).

On average, annual health care expenditures was higher in women than men (€ 4,795 and € 3,814), survivors diagnosed before 1980 (€ 6,970) who likely correspond with survivors between 41–50 years old (€ 7,676), among diagnosed with CNS tumor (€ 7,116), and in FCCSS (€ 4,556 vs € 3,663 in FRANCIM) (Table 1). It is noteworthy that the 167 survivors who died between 2011 and 2016, had an annual mean of expenditures of € 25,611 and accounted for 13.4% of the total expenditures.

### Healthcare utilization

Health care expenditures by items are reported in Table 2. The leading expenditure item was hospitalizations which represents 45% of total expenses and were experienced by 65% of survivors during the study period. The number of hospitalizations was higher in women than in men (73% vs 59%) (S3 Table). The annual mean for hospitalization expenses was € 1,919. Pharmacy was also an important expenditure item representing 16% of total healthcare expenditure with an annual mean of € 676 per patient (€ 719 in men vs 622 in women).

Around half of survivors received sick leave during the 6-year follow-up and their annual mean per patient was € 410 but notably was almost the double for women than for men (€ 557 vs 289) (S3 Table). On the contrary, only 6.5% of patients had expenditures related to disability benefits but their cost reached almost € 8 million which is more than 5% of the total expenses. The distribution of health care expenditure by items was quite similar in both cohorts (S4 Table).

### Health care expenditure by type of primary cancer

Details on expenditures by each type of primary cancer are shown in Table 3. Survivors from CNS tumors had the highest annual mean of hospitalizations expenses (€4,142), while kidney tumor survivors had the highest annual mean of pharmacy (€ 1,578) compared with the other types of cancer. Disability benefits annual mean were higher among bone sarcoma survivors (€ 479) while sick leave where major for thyroid tumor survivors (€ 708).

**Table 1. Survivors characteristics and health care expenditures.**

| | N° Patients | PY | Annual Health care expenditures | |
|---|---|---|---|---|
| | (%) | | Mean (SD) | Median (IQR) |
| Total | 5319 | 31533.6 | 4255 (18790) | 494 (105–2151) |
| Sex | | | | |
| Man | 2929 (55.1) | 17351.9 | 3814 (19289) | 338 (64–1554) |
| Women | 2390 (44.9) | 14181.7 | 4795 (18147) | 721 (198–3290) |
| Year of childhood cancer diagnosis | | | | |
| <1980 | 988 (18.6) | 5760.3 | 6970 (28210) | 995 (242–4552) |
| 1980–1989 | 1852 (34.8) | 10974.5 | 4731 (20024) | 607 (149–2749) |
| > = 1990 | 2479 (46.6) | 14798.8 | 2841 (11813) | 326 (52–1334) |
| Age at childhood cancer | | | | |
| 0–1 | 937 (17.6) | 5575.0 | 2905 (9828) | 257 (15–1337) |
| 2–4 | 1034 (19.4) | 6140.2 | 4926 (26253) | 413 (69–2003) |
| 5–9 | 1088 (20.5) | 6433.3 | 4420 (14300) | 552 (126–2743) |
| 10–14 | 1130 (21.2) | 6685.2 | 5072 (24480) | 640 (170–2686) |
| ≥15 | 1130 (21.2) | 6699.8 | 3790 (12352) | 600 (179–2135) |
| Age at January 2011 (Start date) | | | | |
| <20 | 550 (10.3) | 3291.9 | 1475 (8143) | 0 (0–309) |
| 20–30 | 1979 (37.2) | 11812.8 | 3561 (13189) | 443 (113–1891) |
| 31–40 | 1878 (35.3) | 11142.6 | 4303 (20597) | 592 (157–2312) |
| 41–50 | 753 (14.2) | 4355.4 | 7676 (29734) | 1072 (265–5056) |
| > = 51 | 159 (3) | 930.9 | 6233 (14164) | 1378 (398–5272) |
| French geographical deprivation index** | | | | |
| 1 Quintile | 1063 (20) | 6312.9 | 4569 (24995) | 459 (92–1952) |
| 2 Quintile | 1061 (19.9) | 6293.3 | 3688 (11473) | 507 (118–2020) |
| 3 Quintile | 1066 (20) | 6322.3 | 4034 (16209) | 471 (103–1920) |
| 4 Quintile | 1064 (20) | 6320.5 | 4699 (23783) | 507 (97–2374) |
| 5 Quintile | 1065 (20) | 6284.6 | 4283 (13374) | 539 (119–2529) |
| First primary cancer type | | | | |
| Kidney tumors | 668 (12.6) | 3942.0 | 5021 (30330) | 383 (78–1776) |
| Neuroblastoma | 574 (10.8) | 3434.2 | 2952 (12224) | 285 (37–1301) |
| Lymphoma | 1071 (20.1) | 6350.7 | 3366 (10662) | 442 (106–1770) |
| Soft tissue sarcomas | 523 (9.8) | 3117.3 | 4007 (14074) | 471 (108–2219) |
| Bone sarcomas | 445 (8.4) | 2636.7 | 5207 (14193) | 827 (188–4303) |
| Central nervous system tumor | 756 (14.2) | 4427.7 | 7116 (29617) | 1130 (256–4925) |
| Gonadal tumor | 389 (7.3) | 2322.3 | 3575 (15215) | 392 (93–1401) |
| Thyroid tumor | 109 (2) | 650.7 | 3202 (7882) | 785 (348–2210) |
| Retinoblastoma | 305 (5.7) | 1815.5 | 2852 (9698) | 235 (0–1161) |
| Other solid cancer | 479 (9) | 2836.5 | 3363 (12231) | 495 (143–1797) |
| Cohort | | | | |
| FCCSS | 3589 (67.5) | 21247.6 | 4556 (18830) | 507 (97–2297) |
| French cancer registry | 1730 (32.5) | 10286.0 | 3633 (18692) | 476 (122–1890) |
| Status at December 2016 (Ending date) | | | | |
| Alive | 5152 (96.9) | 30912.0 | 3770 (17677) | 473 (101–2003) |
| Death | 167 (3.1) | 621.6 | 25611 (40950) | 10208 (1437–34703) |

PY: Person-years of follow-up, SD: Standard deviation, IQR: Inter Quartile Range.

**: Ecological Index measuring the deprivation, and based on the median household income, the percentage high school graduates in the population aged 15 years and older, the percentage blue-collar workers in the active population, and the unemployment rate [19].

**Table 2. Health care expenditures by items.**

| | N° Patients (%) | N° Claims | Total Expenditures in Millions € (%) | Annual mean Expenditure Per-Patient in €* (SD) |
|---|---|---|---|---|
| General practitioner visits | 5,055 (95) | 144,107 | 3.1 (2.3) | 97 (133) |
| Other specialists visits | 4,935 (92.8) | 161,998 | 5.4 (4) | 170 (439) |
| Physiotherapy visits | 2,195 (41.3) | 123,828 | 2.1 (1.6) | 67 (319) |
| Nursing visits | 3,519 (66.2) | 118,548 | 1.7 (1.3) | 55 (650) |
| Other health professionals visits † | 597 (11.2) | 12,548 | 0.4 (0.3) | 12 (135) |
| Pharmacy | 5,046 (94.9) | 502,872 | 21.4 (15.9) | 676 (9,736) |
| Medical device | 4,223 (79.4) | 52,479 | 8.2 (6.1) | 260 (1,800) |
| Laboratory test | 4,597 (86.4) | 101,246 | 2.5 (1.9) | 80 (302) |
| Technical medical procedures ‡ | 4,700 (88.4) | 48,583 | 2.6 (1.9) | 83 (287) |
| Transport | 1,743 (32.8) | 24,604 | 4.4 (3.3) | 140 (979) |
| Hospitalizations | 3,450 (64.9) | 35,001 | 60.7 (45.1) | 1,919 (13,730) |
| Disability Benefits § | 345 (6.5) | 13,863 | 7.9 (5.9) | 251 (1,538) |
| Sick Leave | 2,731 (51.3) | 41,722 | 13.0 (9.6) | 410 (1,623) |
| Others | 571 (10.7) | 3,442 | 1.2 (0.9) | 37 (1,039) |
| Total | 5,319 | 1,384,841 | 134.5 | 4,255 (18,790) |

* Annual mean expenditure per-patient in € were calculated for the entire population (= 5,319). † Other medical professional visits included expenditures related to visits to podiatrist, optometrists, speech therapist, and others ‡ Technical medical procedures includes expenditures mainly related to medical imaging techniques. § Disability benefits includes all welfare payments or pensions made by the French Government to assistance people with disabilities.

**Table 3. Annual mean and percentage* (%) of healthcare expenditures items by type of primary cancer.**

| | Kidney tumors (n = 668) | Neuroblastoma (n = 574) | Lymphoma (n = 1071) | Soft tissue sarcomas (n = 523) | Bone sarcomas (n = 445) | Central nervous system tumor (n = 756) | Gonadal tumor (n = 389) | Thyroid tumor (n = 109) | Retinoblastoma (n = 305) | Other solid cancer (n = 479) |
|---|---|---|---|---|---|---|---|---|---|---|
| General practitioner visits | 84 (1.7) | 70 (2.4) | 96 (2.8) | 94 (2.4) | 116 (2.2) | 130 (1.8) | 81 (2.3) | 128 (4) | 72 (2.5) | 102 (3) |
| Other specialist visits | 172 (3.4) | 134 (4.5) | 185 (5.5) | 162 (4.1) | 179 (3.4) | 181 (2.5) | 160 (4.5) | 250 (7.8) | 126 (4.4) | 182 (5.4) |
| Physiotherapy | 37 (0.7) | 35 (1.2) | 49 (1.5) | 55 (1.4) | 90 (1.7) | 182 (2.6) | 37 (1) | 40 (1.2) | 21 (0.7) | 59 (1.8) |
| Nursing visits | 39 (0.8) | 57 (1.9) | 35 (1) | 55 (1.4) | 28 (0.5) | 144 (2) | 29 (0.8) | 22 (0.7) | 25 (0.9) | 56 (1.7) |
| Other health professionals visits | 5 (0.1) | 8 (0.3) | 7 (0.2) | 9 (0.2) | 4 (0.1) | 39 (0.5) | 4 (0.1) | 15 (0.5) | 1 (0) | 16 (0.5) |
| Pharmacy | 1578 (31.4) | 350 (11.8) | 498 (14.8) | 469 (11.7) | 564 (10.8) | 864 (12.1) | 777 (21.7) | 944 (29.5) | 220 (7.7) | 397 (11.8) |
| Medical device | 288 (5.7) | 201 (6.8) | 131 (3.9) | 275 (6.9) | 956 (18.4) | 295 (4.2) | 80 (2.2) | 56 (1.7) | 124 (4.3) | 138 (4.1) |
| Laboratory test | 99 (2) | 63 (2.1) | 85 (2.5) | 67 (1.7) | 80 (1.5) | 87 (1.2) | 74 (2.1) | 130 (4) | 49 (1.7) | 81 (2.4) |
| Technical Medical Procedures | 74 (1.5) | 54 (1.8) | 81 (2.4) | 90 (2.3) | 104 (2) | 93 (1.3) | 101 (2.8) | 110 (3.4) | 57 (2) | 81 (2.4) |
| Transport | 159 (3.2) | 107 (3.6) | 88 (2.6) | 107 (2.7) | 183 (3.5) | 286 (4) | 94 (2.6) | 35 (1.1) | 160 (5.6) | 85 (2.5) |
| Hospitalizations | 1827 (36.4) | 1401 (47.5) | 1345 (40) | 1831 (45.7) | 1863 (35.8) | 4142 (58.2) | 1500 (42) | 658 (20.6) | 1731 (60.7) | 1384 (41.2) |
| Disability Benefits | 245 (4.9) | 152 (5.1) | 276 (8.2) | 287 (7.2) | 479 (9.2) | 303 (4.3) | 205 (5.7) | 91 (2.9) | 69 (2.4) | 176 (5.2) |
| Sick Leave | 380 (7.6) | 309 (10.5) | 460 (13.7) | 482 (12) | 532 (10.2) | 266 (3.7) | 412 (11.5) | 708 (22.1) | 182 (6.4) | 565 (16.8) |
| Others | 34 (0.7) | 12 (0.4) | 30 (0.9) | 23 (0.6) | 30 (0.6) | 103 (1.4) | 19 (0.5) | 15 (0.5) | 15 (0.5) | 39 (1.2) |
| Total | 5 021 € | 2 952 € | 3 366 € | 4 007 € | 5 207 € | 7 116 € | 3 575 € | 3 202 € | 2 852 € | 363 € |

* Percentage (%) of each expenditure items were calculated for each primary cancer population.

## Multivariate analysis of survivor's characteristics impact on healthcare expenditures

Table 4 shows the estimations of the GLM model using gamma distributions after adjusting by patients' characteristics. Total healthcare expenditures were higher at increasing age (Beta: 0.04, p = < .0001), in women (Beta: 0.30, p = < .0001), in patients treated for CNS tumor (Beta: 0.70, as compared to neuroblastoma, p = < .0001), in patients treated between aged 2 and 4 (Beta: 0.36, as compared to age 0–1, p = 0.05) and among survivors who died during 2011 and 2016 (Beta: 1.85, p = < .0001). Annual healthcare expenditures of survivors from FCCSS were not significantly higher than survivors' expenditures from FRANCIM (Beta: 0.12, p = 0.23). These results did not vary by excluding variable "death" from the model or excluding deceased survivors from the study population (Model II and III, respectively).

Fig 1 shows the adjusted estimates for the annual mean of healthcare expenditures from the gamma model by type of primary cancer. Survivors with neuroblastoma, lymphoma, soft tissue sarcoma, gonadal, thyroid tumor and other solid tumors had similar annual mean of health care expenditures between 3,000 and 3,500 euros whereas, survivors with kidney tumor, bone sarcoma and retinoblastoma had a similar mean of expenditures around 4,000 and 4,500 euros per year.

Finally, we investigated each item of the healthcare expenditure separately and showed that women had significantly higher adjusted expenditures than men, for all items, excepted for

**Table 4. Multivariate analysis[*].**

|  | Total Patients (n = 5,319) | | | | Patients Alive (n = 5,152) | |
|---|---|---|---|---|---|---|
|  | Model I | | Model II | | Model III | |
|  | **Beta** | **Pr > \|Z\|** | **Beta** | **Pr > \|Z\|** | **Beta** | **Pr > \|Z\|** |
| Intercept | 11.54 | 0.69 | 1.26 | 0.96 | 21.67 | 0.45 |
| Women | 0.30 | < .0001 | 0.27 | < .0001 | 0.31 | < .0001 |
| Age | 0.04 | < .0001 | 0.06 | < .0001 | 0.04 | < .0001 |
| Year of Diagnosis | 0.00 | 0.85 | 0.00 | 0.86 | -0.01 | 0.59 |
| Age at first cancer (Ref = 0–1) |  |  |  |  |  |  |
| 2–4 | 0.36 | 0.05 | 0.33 | 0.06 | 0.37 | 0.05 |
| 5–9 | 0.08 | 0.64 | 0.07 | 0.69 | 0.10 | 0.56 |
| 10–14 | 0.16 | 0.50 | 0.08 | 0.72 | 0.20 | 0.38 |
| ≥15 | -0.03 | 0.92 | -0.16 | 0.57 | 0.04 | 0.90 |
| French Index Deprivation | 0.02 | 0.47 | 0.04 | 0.27 | 0.03 | 0.40 |
| First primary cancer type (Ref = Neuroblastoma) |  |  |  |  |  |  |
| Kidney tumor | 0.19 | 0.46 | 0.32 | 0.17 | 0.20 | 0.46 |
| Lymphoma | -0.12 | 0.54 | 0.03 | 0.86 | -0.13 | 0.50 |
| Soft tissue sarcoma | -0.03 | 0.87 | 0.07 | 0.71 | -0.04 | 0.83 |
| Bone sarcoma | 0.30 | 0.14 | 0.41 | 0.04 | 0.30 | 0.14 |
| Central nervous system tumor | 0.70 | < .0001 | 0.86 | < .0001 | 0.72 | < .0001 |
| Gonadal tumor | 0.08 | 0.77 | 0.08 | 0.74 | 0.09 | 0.74 |
| Thyroid tumor | 0.01 | 0.98 | 0.01 | 0.98 | 0.02 | 0.95 |
| Retinoblastoma | 0.05 | 0.79 | 0.32 | 0.13 | 0.02 | 0.92 |
| Other solid cancer | -0.05 | 0.81 | 0.11 | 0.59 | -0.05 | 0.80 |
| FCCSS Survivors | 0.12 | 0.23 | 0.14 | 0.16 | 0.12 | 0.22 |
| Death | 1.85 | < .0001 | . | . | . | . |

[*] GLM model using gamma distributions. Model I were adjusted using all variables. Model II excluded "death" variable. Model III excluded dead patients.

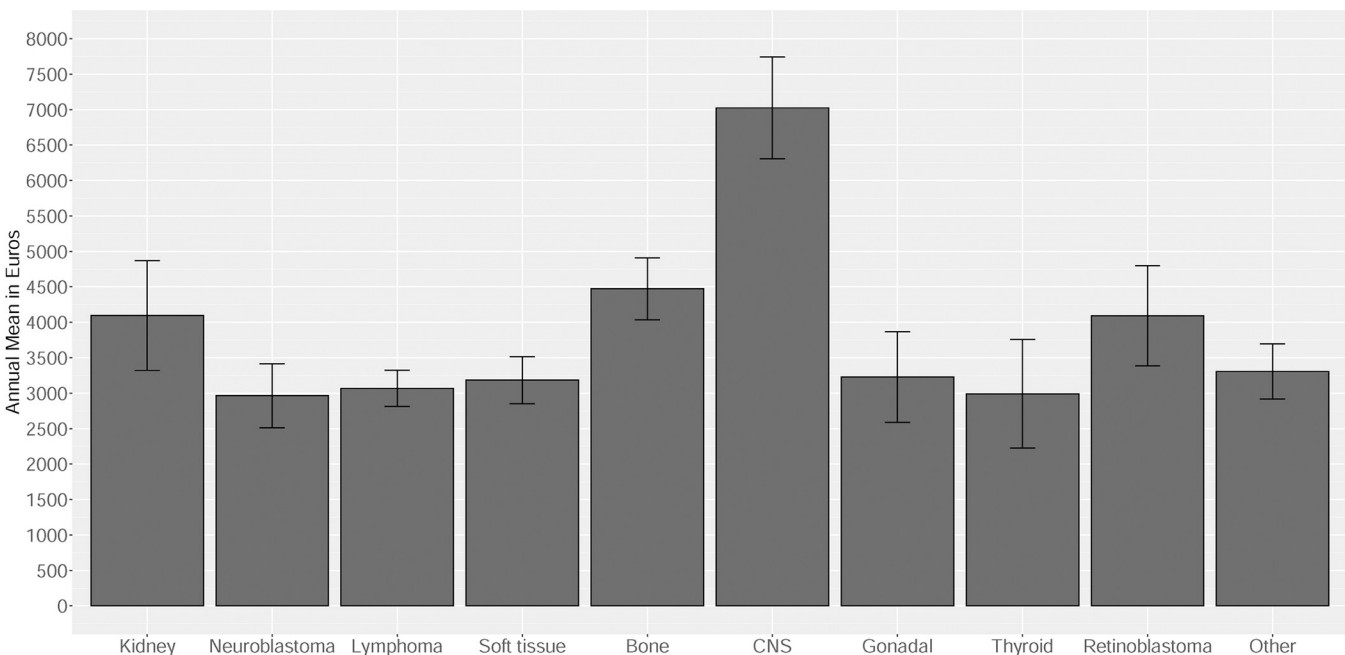

**Fig 1. Adjusted\* annual health care expenditures by type of primary cancer.** \* Adjusted by age, sex, year of diagnosis, age at diagnosis, French index deprivation, cohort and type of primary cancer. Type of primary cancer: Kidney tumor, neuroblastoma, lymphoma, soft tissue sarcoma, bone sarcoma, central nervous system tumor (CNS), gonadal tumor, thyroid tumor, retinoblastoma and other solid cancer.

disability benefits, other health professionals visits, pharmacy, and medical device (S5 Table). Men survivors from FCCSS had higher adjusted expenses than the ones from FRANCIM, this difference was not observed in women (interaction p = <0.001) (S2 Fig). Lastly, no global interaction was shown between the type of primary cancer and the origin of patients, FCCSS or FRANCIM (S3 Fig).

## Discussion

To our knowledge, this is the first detailed study of the economic burden of childhood cancer survivors in France. We found that the annual mean of healthcare expenditures among CCS were € 4,255, which are composed mainly by expenditures on hospitalizations and pharmacy. Additionally, we showed that women had higher expenditures than men, and that CNS tumor survivors had the highest expenditures. Although, FCCSS survivors had higher expenditures than those from FRANCIM, this difference was no longer significant when adjusting on childhood cancer type and on demographics.

CCS have a high rate of illness due to chronic health conditions [21] and require significantly more healthcare resources than the general population [5]. Previous studies have shown that their hospitalization rates are almost twofold increase and their stay was 35% longer than for patients without a cancer history [22]. Consequently, their medical needs translate into substantial healthcare expenditures. In United States, CCS were more likely to have out-of-pocket medical costs [23], and up to 33% of them were unable to see a doctor or go to the hospital due to financial issues [24].

Annual medical expenditures in adolescent or young adult cancer survivors (age 15–35) has been estimated to $7,417 [25], while annual productivity loss among adult survivors of childhood (<14 years at diagnosis) cancer was estimated to $8,169 [8]. In Norway, survivors of cancer at young age have by a four-to fivefold increased risk of not being employed and receive

governmental financial assistance than general population [26]. However, it is important to keep in mind that since the nineties, the rate of iatrogenic events decreases because of the reduction in the use of radiation therapy, and more recently, it could be anticipate that this risk will continue to decrease, in particular because of the emergency of proton-therapy [27].

We also highlighted that a few number of CCS especially survivors who die during the follow up period, were the main expenditure drivers. Most of these deaths have been found related to childhood cancer recurrence during the first two decades and to treatment-related sequelae including cardiovascular diseases and second malignant neoplasms, afterward [28]. The mean expenditures during the last year of life in some types of cancer have been estimated up to 43,000 euros in France [29, 30] which correlates well with our results.

Our findings showing that women survivors have a higher annual mean both in total healthcare expenditures and in several specific expenditures items are in agreement with several studies that have shown that hospitalizations occurred more often among females survivors [31, 32], mostly due to endocrine, metabolic and nutritional disorders and subsequent neoplasms [33]. Additionally all the expenditures associated with pregnancies and perinatal conditions were included in the health expenditures in our analysis. Moreover, women survivors have higher rates of miscarriage or preterm birth than the general population, including risks to both the mother and the fetus [34], which could at least partly explain the difference in the healthcare utilization compared with men survivors.

As expected, expenditures were also higher in CNS tumor survivors. A previous study showed that the cumulative burden of chronic health conditions at age 50 was higher in CCS with CNS tumor than in CCS with any other cancer [35]. This, together with the evidence that survivors from CNS tumor have at least one disability condition [36], were less likely to progress in educational attainment [37] and to have a higher risk of unemployment and reduced incomes compared with the cancer-free population [26, 38], explained their excess of healthcare expenditures. Additionally, progressive disease or relapses more than 5 years after diagnosis of a brain tumor in children is common, particularly, in slowly evolving low grade tumors [39].

Despite the fact that FCCSS survivors were treated in specialized centers to treat cancer, and thus could have been more adverse cases or received a more intense treatment, no significant difference in expenses were found at long term when adjusting on demographical factors and type of childhood cancer to others CCS in France. Although, survivors from FCCSS were younger at time of diagnosis and were recruited in early years than survivors from French cancer registries, they weren't particularly older during follow-up period.

An advantage of our study is to have worked on a large sample of long term survivors. Also, we used a national administrative database which allowed accounting for comprehensive health care expenditures during six years using two sources of long-term CCS in France. Another strength is the inclusion period starting in 1945, which allows us to study variations in cost across the age spectrum.

However, our study is subject to some limitations. First of all, data for a cancer-free control group were not available, which limited our results to the CCS population. Secondly, FRANCIM is a network of population-based registries which does not have national coverage. Thirdly, we were unable to address the association between cancer treatments received by survivors and health expenditures, due to the lack of therapeutic information in survivors from FRANCIM. All in all, our findings provide a first estimation on annual expenditures and the economic burden of CCS in France by type of childhood cancer, and demographical characteristics of survivors. Additionally, transferability to other contexts may be limited to the France territory.

## Conclusion

We have estimated and described the magnitude of health expenditures related to consequences in adulthood of having had cancer treated in childhood. These high expenditures in relation to the age of survivors are related to the more frequent multimorbidity than in the general population. These results lead us to recommend that special attention be paid to this population, particularly in terms of prevention of complications and early medical follow-up. Future research should focus on addressing in deep the relationship between cancer treatment and future healthcare expenditures to establish long-term cost-effectiveness of childhood cancer treatment.

## Supporting information

**S1 Table. Participating French administrative areas in FRANCIM.**
(DOCX)

**S2 Table. Survivor's characteristics by cohort.**
(DOCX)

**S3 Table. Health care expenditures by sex.**
(DOCX)

**S4 Table. Health care expenditures by cohort.**
(DOCX)

**S5 Table. Multivariate analysis for each type of expenditure.**
(DOCX)

**S6 Table. Multivariate analysis for total health care expenditure by each type of cancer.**
(DOCX)

**S1 Fig. Histogram of the survivors mean of healthcare expenditures.**
(TIF)

**S2 Fig. Adjusted annual health care expenditures by sex and cohort.**
(TIF)

**S3 Fig. Adjusted annual health care expenditures by type of primary cancer and cohort.**
(TIF)

## Acknowledgments

The authors thank the staff of each member of the FRANCIM network who participated in the collection of data; Claire Schvartz (Registre des Cancers Thyroidiens de la Marne et des Ardennes), Michel Velten (Registre des Cancers du Bas-Rhin), Anne-Valérie Guizard (Registre Général des Tumeurs du Calvados), Guy Launoy (Registre des tumeurs digestives du Calvados), Anne-Marie Bouvier (Registre Bourguignon des cancers Digestifs), Anne-Sophie Woronoff (Registre des tumeurs du Doubs), Karima Hammas (Registre des tumeurs du Haut-Rhin), Brigitte Trétarre (Registre des Tumeurs de l'Hérault), Marc Colonna (Registre du Cancers de l'Isère), Brigitte Lacour (Registre National des Tumeurs Solides de l'Enfant), Simona Bara (Registre des Cancers de la Manche), Clarisse Joachim (Registre de la Martinique), Bénédicte Lapôtre-Ledoux (Registre de la Somme), Pascale Grosclaude (Registre des Cancers du Tarn), and Florence Molinié (Registre de la Loire-Atlantique et Vendée).

## Author Contributions

**Conceptualization:** Nathalie Pelletier-Fleury, Florent de Vathaire.

**Data curation:** Daniel Bejarano-Quisoboni, Julie Nys, Angela Jackson, Giao Vu-Bezin.

**Formal analysis:** Daniel Bejarano-Quisoboni, Florent de Vathaire.

**Investigation:** Daniel Bejarano-Quisoboni, Nathalie Pelletier-Fleury, Brigitte Lacour, Pascale GrosClaude, Agnès Dumas, Brice Fresneau, Florent de Vathaire.

**Methodology:** Daniel Bejarano-Quisoboni, Nathalie Pelletier-Fleury, Brice Fresneau, Florent de Vathaire.

**Project administration:** Brigitte Lacour, Pascale GrosClaude, Julie Nys, Florent de Vathaire.

**Supervision:** Nathalie Pelletier-Fleury, Florent de Vathaire.

**Validation:** Daniel Bejarano-Quisoboni, Rodrigue S. Allodji, Brigitte Lacour, Pascale GrosClaude, François Doz, Delphine Berchery, Claire Pluchart, Piere-Yves Bondiau, Julie Nys, Angela Jackson, Charlotte Demoor-Goldschmidt, Agnès Dumas, Cécile Thomas-Teinturier, Nadia Haddy, Brice Fresneau, Florent de Vathaire.

**Writing – original draft:** Daniel Bejarano-Quisoboni.

**Writing – review & editing:** Daniel Bejarano-Quisoboni, Rodrigue S. Allodji, Brigitte Lacour, Pascale GrosClaude, Hélène Pacquement, François Doz, Delphine Berchery, Claire Pluchart, Piere-Yves Bondiau, Julie Nys, Angela Jackson, Charlotte Demoor-Goldschmidt, Agnès Dumas, Cécile Thomas-Teinturier, Giao Vu-Bezin, Dominique Valteau-Couanet, Nadia Haddy, Brice Fresneau, Florent de Vathaire.

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
