## [Decision Letter · Decision Letter 0]

13 Jan 2022

PONE-D-21-17399

Health care expenditures among long-term childhood cancer survivors in France

PLOS ONE

Dear Dr. de Vathaire,

Thank you for submitting your manuscript to PLOS ONE. After careful consideration, we feel that it has merit but does not fully meet PLOS ONE’s publication criteria as it currently stands. Therefore, we invite you to submit a revised version of the manuscript that addresses the points raised during the review process.

The manuscript has been evaluated by two reviewers, and their comments are available below.

The reviewers have raised a number of major concerns. They feel that the title of the article should clarify the sub-population of childhood cancer survivors under investigation, and the Introduction and the Discussion should clarify the exclusion of non-cancer comparison group. They request improvements to the reporting of methodological aspects of the study, for example, regarding the data source. The reviewers also note concerns about the language and request copy-editing.

Could you please carefully revise the manuscript to address all comments raised?

We look forward to receiving your revised manuscript.

Kind regards,

Lorena Verduci

Staff Editor

PLOS ONE

Journal Requirements:

5. One of the noted authors is a group or consortium FRANCIM-Group. In addition to naming the author group, please list the individual authors and affiliations within this group in the acknowledgments section of your manuscript. Please also indicate clearly a lead author for this group along with a contact email address.

Reviewers' comments:

Reviewer's Responses to Questions

**Comments to the Author**

1. Is the manuscript technically sound, and do the data support the conclusions?

Reviewer #1: Yes

Reviewer #2: Yes

2. Has the statistical analysis been performed appropriately and rigorously? 

Reviewer #1: Yes

Reviewer #2: Yes

3. Have the authors made all data underlying the findings in their manuscript fully available?

Reviewer #1: Yes

Reviewer #2: Yes

4. Is the manuscript presented in an intelligible fashion and written in standard English?

Reviewer #1: Yes

Reviewer #2: Yes

5. Review Comments to the Author

Reviewer #1: The authors present an analysis that examines healthcare expenditure from 2011-2016 among childhood cancer survivors in France diagnosed between 1945 and 2000. Financial toxicity and cost of health care expenditure among childhood cancer survivors is very topical. The manuscript is well written and generally easy to follow. There are some grammatical issues that will need to be addressed during copy-editing. I have provided some comments for the authors to consider which are provided in no particular order.

- The title of the article needs to indicate the sub-population of childhood cancer survivors under investigation.

- The second sentence of the introduction describes an observed survival proportion, but the authors refer to this as a rate. Please remove the word ‘rates’ from this sentence.

- In the third paragraph of the introduction, the authors describe the data source (FCCSS). I think this information belongs in the methods section where the data source should be described. At the end of this paragraph the authors hypothesize about a potential bias associated with the FCCSS. I wonder if there are any references that could be provided to highlight this? I also feel the authors need to consider this potential bias in the limitations section more fully specifically with regard to how to interpret the results presented in this analysis.

- In the study population section, the authors indicate that the rates of linkage to the SNDS is 55.6% and 71.9% for each of the two data sources. Given this differential linkage and reasonably low rates of linkage there is opportunity for bias. The authors need to spend a little more time examining the factors that drive the variability in linkage. I would suggest a simple logistic regression analysis where the outcome is linked and various factors from table 1 can be included. This analysis may need to be stratified by data source to determine if the factors affecting linkage are different. The full results of this analysis need not be included in the manuscript, but can form the basis for a fulsome discussion about the potential bias, how it might be operating and the potential impact on the analysis presented.

- It took me some time to consider the inclusion of the diagnoses in the early years (they start in 1945) given the observation window for the outcome only starts in 2011, but I think this approach is strong as it produces estimates across the age spectrum.

- In the Primary Measures section, the authors indicate that the expenditures are expressed in real terms using the CPI. The authors need to be explicit in terms of the anchor year for the adjustment and provide a reference to the CPI data that was used for the adjustment.

- Given the longitudinal nature of the outcome ascertainment, I think it would strengthen the paper to provide the person-years of follow-up in all of the tables.

- Given what appears to be skewed nature of the expenditures, it might be more appropriate to provide the median and IQR or both the mean and SD and median and IQR.

- There are two factors that likely drive 2 of the main results. The first is the higher costs among female survivors and the second is how death is related to higher costs. Neither of these are particularly surprising given births among female survivors are known to considerably contribute to health care costs. Is it possible for these costs to be removed for a sensitivity analysis so this could be examined? Further, could costs in the last period of life among those that die, say 6 or 8 weeks, be excluded so as to assess the timing of these costs relative to death?

- In table 2 I struggle to make sense of the data presented. For example, consider paramedic, the annual mean expenditure per patient is presented as 12 so over the 6 year period of outcome assessment it would be 72. If I multiply this by the number of patients, 597, the total is 42,984 which is very different (by a factor of 10) than the 400,000 reported in the table. Why is this? Am I just miss-interpreting the table?

- It would be good to include column percentages in table 3 so a reader could compare across groups.

- Please include more explicit column headings or notes to describe in the table. It is difficult to understand the table in the absence of reading the manuscript.

Reviewer #2: Thank you for the opportunity to review this manuscript. This is a population-based study of financial expenditures in registry-defined cohorts of childhood cancer survivors in France. The authors have documented the mean annual health system costs (i.e., excluding out of pocket expenses, etc.) from 2011-2016. The highest expenses were incurred for survivors of CNS tumors, survivors who were women and older during the study period. The data submitted are complete, statistical analyses correct, conclusions appropriate to results, and the paper is generally well-written.

Please consider the following questions and comments, which are offered in hopes of strengthening this contribution:

1. Would the authors please provide an explanation in the Introduction and/or Discussion why they did not include a non-cancer comparison group matched for age, sex, and age at diagnosis? This would provide some perspective for these results, which are limited to the childhood cancer population. Also, this would help elucidate what role "normal" health care costs played, such as pregnancy in female survivors.

1a. The analysis does include the two oncology cohorts, one from the FCCSS and the other from FRANCIM. A more complete explanation of how these two sources complement each other in interpreting the data would enhance the conclusions (i.e., what were authors hoping to discover by comparing these two cohorts?).

2. Would the authors please explain why this sample was limited to solid tumors and excluded leukemias.

3. For the expenditures listed in Table 2, it would be helpful to define these terms/categories, as some are not self-evident. For example, what are "sick payments"-? Presumably these terms are defined within the French system but the terms may be ambiguous for other national systems.

4. Page 14, last paragraph, next to last line: a typographical error ("where" and should be "were").

5. For the regression analysis on tumor type, can the authors please explain in the manuscript why neuroblastoma was chosen as the referent?

6. Page 19, second paragraph, the sentence beginning, "Studies from the United States..." is difficult to understand. Are there some words missing? It is informative, but also rather long and hard to follow. Consider clarifying the language and also splitting it up into at least 2 smaller sentences.

7. In the Introduction, the authors suggest the analysis would address potential differences between patients cared for in the private vs. public health systems. Was this reported in the Results?

8. Reference 27 does not have the year of publication.

6. PLOS authors have the option to publish the peer review history of their article (what does this mean?). If published, this will include your full peer review and any attached files.

Reviewer #1: No

Reviewer #2: No

---

## [Author Response · Author response to Decision Letter 0]

25 Feb 2022

Dear Editor-in-chief, 

We are grateful to the editors and reviewers for their time and constructive comments on our manuscript. We have implemented their suggestions and answered their concerns which, we believe, allowed improving the manuscript. Changes in the initial version of the manuscript can be retrieved in the tracked change version. Below, we provide a point-by-point response explaining how we have addressed each of the editors or reviewers’ comments. We look forward to receiving your further evaluation of our manuscript.

Sincerely,

Florent de Vathaire, PhD

Head of the Radiation Epidemiology Group

Unit 1018 INSERM – CESP

Institut Gustave Roussy

39, rue Camille Desmoulins 

94805 Villejuif, France

Tel: +33 1 42 11 54 57

Fax: +33 1 42 11 53 15

Email: florent.devathaire@gustaveroussy.fr

Web: https://www.gustaveroussy.fr/ - https://cesp.inserm.fr/fr

All page and line numbering below refers to the re-submitted clean copy, word document, which has been line numbered, before re-submission.

Journal Requirements:

Our response: We thank the editor for your review. We have made several changes in order to fully meet PLOS ONE's style requirements.

Our response: We have added more details regarding participant consent.

Lines 133-137 “The study was approved by the French Data Protection Authority (CNIL) (Authorization n°902287) and by the ethics committee of the INSERM. Patient informed consent was not required for this study because we obtained a specific act in law from the French “Conseil d’Etat”, the highest court in France (Order 2014-96 of 2014 February 3), that approved the cession of the SNDS data for all patients included in the FCCSS and FRANCIM.”

Our response: We have added more details regarding grant information of DBQ. However, for the other of grants or funding that we received we don’t have a grants numbers because the funders didn’t provide. 

DBQ received a doctoral grant from the Paris-Saclay University (N° contract 9R_2019_PSU000101141_Upsud). This work was funded by the Ligue Nationale Contre le Cancer (Grant N°RAB20035LLA). The FCCSS cohort was funded by the French Society of Cancer in Children and adolescents (SFCE), the Gustave Roussy Fondation, the Foundation ARC (POPHarC program) and The French National Research Agency (ANR, HOPE-EPI project). The funders had no role in study design, data collection and analysis, decision to publish, or preparation of the manuscript.

Our response: We have linked the ORCID account for the corresponding author, Florent de Vathaire (0000-0002-8374-9281).

5. One of the noted authors is a group or consortium FRANCIM-Group. In addition to naming the author group, please list the individual authors and affiliations within this group in the acknowledgments section of your manuscript. Please also indicate clearly a lead author for this group along with a contact email address.

Our response: We added individual authors and affiliations of FRANCIM-Group in the acknowledgments section. The Lead author for this group is Florence Molinié (florence.molinie@chu-nantes.fr).

Lines 378-389 “The authors thank the staff of each member of the FRANCIM network who participated in the collection of data; Claire Schvartz (Registre des Cancers Thyroidiens de la Marne et des Ardennes), Michel Velten (Registre des Cancers du Bas-Rhin), Anne-Valérie Guizard (Registre Général des Tumeurs du Calvados), Guy Launoy (Registre des tumeurs digestives du Calvados), Anne-Marie Bouvier (Registre Bourguignon des cancers Digestifs), Anne-Sophie Woronoff (Registre des tumeurs du Doubs), Karima Hammas (Registre des tumeurs du Haut-Rhin), Brigitte Trétarre (Registre des Tumeurs de l'Hérault), Marc Colonna (Registre du Cancers de l'Isère), Brigitte Lacour (Registre National des Tumeurs Solides de l'Enfant), Simona Bara (Registre des Cancers de la Manche), Clarisse Joachim (Registre de la Martinique), Bénédicte Lapôtre-Ledoux (Registre de la Somme), Pascale Grosclaude (Registre des Cancers du Tarn), and Florence Molinié (Registre de la Loire-Atlantique et Vendée)”. 

Reviewers' Comments to Author:

Reviewer #1:

The authors present an analysis that examines healthcare expenditure from 2011-2016 among childhood cancer survivors in France diagnosed between 1945 and 2000. Financial toxicity and cost of health care expenditure among childhood cancer survivors is very topical. The manuscript is well written and generally easy to follow. There are some grammatical issues that will need to be addressed during copy-editing. I have provided some comments for the authors to consider which are provided in no particular order.

Our response: Thank you very much for your review.

1. The title of the article needs to indicate the sub-population of childhood cancer survivors under investigation.

Our response: Taking up your suggestion, we have changed the title of the article to: “Health care expenditures among long-term solid childhood cancer survivors in France: Results from the French Childhood Cancer Survivor Study (FCCSS) and the French network of cancer registries (FRANCIM)”.

2. The second sentence of the introduction describes an observed survival proportion, but the authors refer to this as a rate. Please remove the word ‘rates’ from this sentence.

Our response: The word “rate” has been removed (Line 69). 

3. In the third paragraph of the introduction, the authors describe the data source (FCCSS). I think this information belongs in the methods section where the data source should be described. At the end of this paragraph the authors hypothesize about a potential bias associated with the FCCSS. I wonder if there are any references that could be provided to highlight this?. I also feel the authors need to consider this potential bias in the limitations section more fully specifically with regard to how to interpret the results presented in this analysis.

Our response: Thank you very much for this comment.

Indeed as said by the reviewer, we have hypothesized about the potential bias associated with the FCCSS because these patients were treated in Centers for the fight against cancer in France (French acronym: CLCC). These centers are dedicated to cancer treatment so we believe that they treat the most serious cancer patients or can provide a different kind of health care especially in pediatric patients (like survivors of the FCCSS) which would impact long-term patient outcomes and therefore be reflected in future costs. Our hypothesis then was that the FCCSS survivors would be patients with higher health care costs than other childhood cancer survivors treated in other facilities, because either they were more severe cases or they could have received more intense treatment that would leave more serious long-term late effects. 

However, until now we do not have any reference that could highlight this hypothesis, and one of the aim of this study was to investigate this issue. It is why we provide some details about the data source of the FCCSS in the introduction. We have made adjustments in the introduction to clarify this idea.

Lines 87-95 “The French Childhood Cancer Survivor Study (FCCSS) which includes CCS treated before 2001 provides an opportunity to detail the overall healthcare expenditures among very long-term cancer survivors. Nevertheless, FCCSS survivors were treated during their childhood in Centers for the fight against cancer (CLCC) which are specialized hospitals in cancer treatment in France. Therefore, we hypothesize that the FCCSS could include patients with more advanced and/or aggressive cancers, or they may have received more innovative treatments and consequently differ in terms of long-term outcomes and future health expenses from other CCS treated in other hospital settings.

4. In the study population section, the authors indicate that the rates of linkage to the SNDS is 55.6% and 71.9% for each of the two data sources. Given this differential linkage and reasonably low rates of linkage there is opportunity for bias. The authors need to spend a little more time examining the factors that drive the variability in linkage. I would suggest a simple logistic regression analysis where the outcome is linked and various factors from table 1 can be included. This analysis may need to be stratified by data source to determine if the factors affecting linkage are different. The full results of this analysis need not be included in the manuscript, but can form the basis for a fulsome discussion about the potential bias, how it might be operating and the potential impact on the analysis presented.

Our response: We thank you for this comment. 

Indeed, the high rate of failure in the linkage with SNDS is an issue and it is necessary to deeply investigate its reasons. 

As you suggested we conducted a logistic regression analysis using the all factors from table 1 (excepted FDEP which was available only for patients successfully linked), followed by a stepwise to examine the statistical significance of these factors. The final model in the case of the FCCSS shows that the only factors that drive the linkage are actually the age in 2006 (OR: 1.014 95% CI 1.010 - 1.019) where at increasing age, patients were more likely to be linked; and death (OR: 0.392 95% CI 0.329 - 0.466) where the dead patients were less likely to be linked. For FRANCIM patients, only death was significant (OR: 0.472 95% CI 0.310 - 0.717). We did not find any role of the following factors: type of cancer or of the gender, age at diagnosis.

This means that in both population patients, the identification variables (family and first name, sex, date and place of birth) included probably more error in death patients that in the other ones. Indeed, before 2000, most of this information was not computerized and the errors had more chances to be corrected, in routine, if the patients went frequently to the hospital, but no computerized information was transmitted to the hospital in case of death. This reason could also explains the role of age in 2006 in the FCCSS, which contains patients treated since to 1945, i.e. more long time ago as compared to the ones of FRANCIM Network: older the patient in 2006, the longer his has been treated, and more frequent were the errors in registrations .

5. It took me some time to consider the inclusion of the diagnoses in the early years (they start in 1945) given the observation window for the outcome only starts in 2011, but I think this approach is strong as it produces estimates across the age spectrum.

Our response: We agree it is a strength of this FCCSS study that it allows an analysis across the age spectrum.

6. In the Primary Measures section, the authors indicate that the expenditures are expressed in real terms using the CPI. The authors need to be explicit in terms of the anchor year for the adjustment and provide a reference to the CPI data that was used for the adjustment.

Our response: We have added more details on this point.

Lines 169-171 “All expenditures were expressed in real terms using the consumer price index with a 2015 base year provided by the National Institute of Statistics and Economic Studies (INSEE).”

7. Given the longitudinal nature of the outcome ascertainment, I think it would strengthen the paper to provide the person-years of follow-up in all of the tables.

Our response: Taking up your suggestion, we added Person-Year columns in Table 1 and Table S1 where patients’ characteristics are showed.

8. Given what appears to be skewed nature of the expenditures, it might be more appropriate to provide the median and IQR or both the mean and SD and median and IQR.

Our response: We have added the median and IQR in Table 1. 

9. There are two factors that likely drive 2 of the main results. The first is the higher costs among female survivors and the second is how death is related to higher costs. Neither of these are particularly surprising given births among female survivors are known to considerably contribute to health care costs. Is it possible for these costs to be removed for a sensitivity analysis so this could be examined? Further, could costs in the last period of life among those that die, say 6 or 8 weeks, be excluded so as to assess the timing of these costs relative to death?.

Our response: We thank you for your suggestion. 

We agree about the high impact of death on costs. This is why we have performed 3 models in our multivariate analysis. One model included the death variable, the second one excluded it and the third model excluded patients who died (i.e. correspond to the sensitivity analysis required by the reviewer), in order to be able to appreciate the stability of our results (Table 4). In the first model (n=5319), we observed that "death" is an explanatory factor for expenditures (beta=1.85, p<0.0001), however, when this variable is removed from the model (model 2, n=5319) or when the model is run only on living people (n=5152), the estimators remain almost identical. So, we believe that dying does not seem to be a major confounding factor in the models

Regarding pregnancy and childbirth, it seems very difficult to disentangle all pregnancy-related expenses because this would involve an exhaustive search of consultations, maternity leaves, medical procedures, drug codes, causes of hospitalization, in all the different information systems that are coded using different nomenclatures (ICD10, ATC Codes, medical procedures of French national health insurance, etc.).

In order to reply to the questioning of the reviewer, we performed the following analysis: We have identified women who have been pregnant at least once, during study follow-up period, and we have re-estimated the models presented in Table 4, adding the dichotomous variable "pregnancy" yes/no, in order to evaluate the impact of this variable together with the sex variable. The results did not indicate that being pregnant was associated to higher costs (Coef: 0.0205 p-value: 0.8117) neither in the main model nor in the others models. This surprising could be due to the fact the costs of pregnancies could be counterbalanced by the fact that women who were pregnant were in a better health status (thus less expenditures) than the others. 

10. In table 2 I struggle to make sense of the data presented. For example, consider paramedic, the annual mean expenditure per patient is presented as 12 so over the 6 year period of outcome assessment it would be 72. If I multiply this by the number of patients, 597, the total is 42,984 which is very different (by a factor of 10) than the 400,000 reported in the table. Why is this? Am I just miss-interpreting the table?.

Our response: As you had pointed out there is a slight misunderstanding. 

Taking back your example, the annual mean expenditure per patient for Paramedic care is 12. This has been calculated as the division of the total expenditure divided by the number of years and divided by the total number of survivors (n=5319) and not only for those who have spent in paramedic care (n=597).

The exact numbers for this example are 370,325 € / 6 / 5319 which is 11.6 and is rounded to 12 € in the tables presented. If we calculate the average expenditure for only the patients who spent on paramedic care this value rise up to 103,4 € which is very different and by almost a factor of 10. 

We believe that the important values to show the readers is the annual mean taking into account all survivors. Therefore, all figures in the column "Annual mean Per-Patient in €" correspond to the mean of all patients included in the study (n=5319). 

In order to do easier the understanding of the article, we added this information in a foot note of the table 2. 

Line 234 * Annual mean expenditure per-patient in € were calculated for the entire population (=5,319).

11. It would be good to include column percentages in table 3 so a reader could compare across groups.

Our response: We have added the percentages in Table 3. 

12. Please include more explicit column headings or notes to describe in the table. It is difficult to understand the table in the absence of reading the manuscript.

Our response: We have added some foot notes in the tables in order to make them clearer.

Reviewer #2:

Thank you for the opportunity to review this manuscript. This is a population-based study of financial expenditures in registry-defined cohorts of childhood cancer survivors in France. The authors have documented the mean annual health system costs (i.e., excluding out of pocket expenses, etc.) from 2011-2016. The highest expenses were incurred for survivors of CNS tumors, survivors who were women and older during the study period. The data submitted are complete, statistical analyses correct, conclusions appropriate to results, and the paper is generally well-written.

Please consider the following questions and comments, which are offered in hopes of strengthening this contribution:

Our response: Thank you very much for your review and your helpful comments.

1. Would the authors please provide an explanation in the Introduction and/or Discussion why they did not include a non-cancer comparison group matched for age, sex, and age at diagnosis? This would provide some perspective for these results, which are limited to the childhood cancer population. Also, this would help elucidate what role "normal" health care costs played, such as pregnancy in female survivors.

Our response: We fully agree that an analysis with a non-cancer comparison group would help to complete the results provided in this paper. However, at the time we carried out this study we did not have access to the data for the general French population or any other non-cancer comparison group. The processes to obtain this kind of data in France are slow and have been further delayed by COVID.

Therefore, we could not take into account a non-cancer comparison group in our analysis. We have added this as a limitation of our study:

Line 358-359 “However, our study is subject to some limitations. First of all, data on a comparison group without cancer were not available which limited our results to the CCS population.

1a. The analysis does include the two oncology cohorts, one from the FCCSS and the other from FRANCIM. A more complete explanation of how these two sources complement each other in interpreting the data would enhance the conclusions (i.e., what were authors hoping to discover by comparing these two cohorts?).

Our response: Thank you very much for this comment.

Indeed as said by the reviewer, we have hypothesized about the potential bias associated with the FCCSS because these patients were treated in Centers for the fight against cancer in France (French acronym: CLCC). These centers are dedicated to cancer treatment so we believe that they treat the most serious cancer patients or can provide a different kind of health care especially in pediatric patients (like survivors of the FCCSS) which would impact long-term patient outcomes and therefore be reflected in future costs. Our hypothesis then was that the FCCSS survivors would be patients with higher health care costs than other childhood cancer survivors treated in other facilities, because either they were more severe cases or they could have received more intense treatment that would leave more serious long-term late effects. We have made adjustments in the introduction to clarify this idea.

Lines 87-95 “The French Childhood Cancer Survivor Study (FCCSS) which includes CCS treated before 2001 provides an opportunity to detail the overall healthcare expenditures among very long-term cancer survivors. Nevertheless, FCCSS survivors were treated during their childhood in Centers for the fight against cancer (CLCC) which are specialized hospitals in cancer treatment in France. Therefore, we hypothesize that the FCCSS could include patients with more advanced and/or aggressive cancers, or they may have received more innovative treatments and consequently differ in terms of long-term outcomes and future health expenses from other CCS treated in other hospital settings.”

2. Would the authors please explain why this sample was limited to solid tumors and excluded leukemias.

Our response: The French Childhood Cancer Survivor Study (FCCSS) is a cohort built by the Radiation Epidemiology Team of CESP between 1985 and 1995 and concerns only people treated for a solid tumor or lymphoma. In France there is another cohort called The French Childhood Cancer Survivor Study for Leukemia (LEA) for survivors treated for leukemia which is coordinated by another group.

To make this clear from the beginning we have stated in the title, abstract and study population section that is a study for solid cancer survivors.

Line 2-4 “Health care expenditures among long-term solid childhood cancer survivors in France: Results from the French Childhood Cancer Survivor Study (FCCSS) and the French network of cancer registries (FRANCIM)”.

Line 51-53 “A total of 5319 five-year solid CCS diagnosed before the age of 21 between 1945 and 2000 in France were identified in the French Childhood Cancer Survivors Study cohort (FCCSS) and the French cancer registry".

Lines 104-105 “The FCCSS is a retrospective cohort of 7,670 five-year CCS diagnosed for solid cancer or lymphoma (all malignancies except leukemia)".

Additionally, we added the following lines in the study population sections:

Line 112-113 “Leukemia survivors from FRANCIM were excluded for better comparability with the FCCSS which did not include leukemia”.

3. For the expenditures listed in Table 2, it would be helpful to define these terms/categories, as some are not self-evident. For example, what are "sick payments"-? Presumably these terms are defined within the French system but the terms may be ambiguous for other national systems.

Our response: We thank you for pointing out that some categories were not clear enough. We have changed the names of the following categories for a more accurate translation to make them clearer:

 Family medicine to “General practitioner visits”

 Medical Specialty to “Other specialist visits” 

 Kinesiotherapy to “Physiotherapy visits”

Nursing to “Nursing visits”

Paramedic to “Other paramedic visits”

Laboratory to “Laboratory test” 

Sick payments to “Sick leaves”. 

We have also added a footnote to better define “Technical Medical Procedures” and "Disability Benefits" in the Table 2:

Lines 234-237 “**Technical medical procedures includes expenditures mainly related to medical imaging techniques. *** Disability benefits includes all welfare payments or pensions made by the French Government to assistance people with disabilities”.

4. Page 14, last paragraph, next to last line: a typographical error ("where" and should be "were").

Our response: We thank you for pointing out the mistake, the word has been changed. 

5. For the regression analysis on tumor type, can the authors please explain in the manuscript why neuroblastoma was chosen as the referent?

Our response: We have added more details in the statistical analysis section for the referent.

Lines 191-193 “Neuroblastoma was chosen as the referent for type of primary cancer variable since was the most homogeneous group in medically terms."

6. Page 19, second paragraph, the sentence beginning, "Studies from the United States..." is difficult to understand. Are there some words missing? It is informative, but also rather long and hard to follow. Consider clarifying the language and also splitting it up into at least 2 smaller sentences.

Our response: We agree that the wording was previously misleading. We clarified this as follows:

Lines 310-315 “In the United States, CCS were more likely to have out-of-pocket medical costs [23], and up to 33% of them were unable to see a doctor or go to the hospital due to financial issues [24]. 

Annual medical expenditures in adolescent or young adult cancer survivors (age 15-35) has been estimated to $7,417 [25], while annual productivity loss among adult survivors of childhood (<14 years at diagnosis) cancer was estimated to $8,169 [8].”

7. In the Introduction, the authors suggest the analysis would address potential differences between patients cared for in the private vs. public health systems. Was this reported in the Results?

Our response: We think there was a misunderstanding. 

The FCCSS cohort included patients treated in Centers for the fight against cancer (French acronym: CLCC) that are private non-profit health establishments of a university-hospital nature participating in the public hospital service in France. That means, they are financed by French health insurance and are controlled by the Ministry of Health, under the same conditions as public hospitals.

Therefore we would not address the differences between patients cared for in the private vs. public hospital since all patients were treated in public establishments. We removed the words “private” and “public” from the introduction, and we clarified this as follows:

Lines 90-91 “Nevertheless, FCCSS survivors were treated during their childhood in Centers for the fight against cancer (CLCC) which are specialized hospitals in cancer treatment in France.” ….. 

Line 94-95 “from other CCS treated in other hospital settings.”

8. Reference 27 does not have the year of publication.

Our response: We thank you for pointing out the mistake, the reference has been updated. 

Lines 483-485 “27. Indelicato DJ, Bates JE, Mailhot Vega RB, Rotondo RL, Hoppe BS, Morris CG, et al. Second tumor risk in children treated with proton therapy. Pediatr Blood Cancer. 2021;68: e28941. doi:10.1002/pbc.28941"

---

## [Editor Report · Decision Letter 1]

16 Mar 2022

PONE-D-21-17399R1Health care expenditures among long-term solid childhood cancer survivors in France: Results from the French Childhood Cancer Survivor Study (FCCSS) and the French network of cancer registries (FRANCIM)PLOS ONE

Dear Dr. de Vathaire,

Thank you for submitting your manuscript to PLOS ONE. After careful consideration, we feel that it has merit but does not fully meet PLOS ONE’s publication criteria as it currently stands. Therefore, we invite you to submit a revised version of the manuscript that addresses the points raised during the review process.

We look forward to receiving your revised manuscript.

Kind regards,

David R Freyer

Academic Editor

PLOS ONE

Journal Requirements:

Additional Editor Comments:

Thank you for submitting your revised manuscript. Please consider the following additional comments in an additional revision and resubmission.

1. The wording of the revised title is awkward. Consider, "Healthcare expenditures among long-term survivors of pediatric solid tumors: Results from...."

2. The response to Comment 5 of Reviewer 1 should include the addition of this strength of the FCCSS study (long follow-up) to the Discussion.

3. The response to Comment 5 of Reviewer 2 is not clear and needs further explanation, including what is meant by the neuroblastoma group being "medically homogenous."

4. The term "paramedic" referring to costs needs clarification/definition.
---

## [Author Response · Author response to Decision Letter 1]

5 Apr 2022

Dear Editor-in-chief, 

We are grateful to the editors and reviewers for their time and constructive comments on our manuscript. We have implemented their suggestions and answered their concerns which, we believe, allowed improving the manuscript. Changes in the initial version of the manuscript can be retrieved in the tracked change version. Below, we provide a point-by-point response explaining how we have addressed each of the editors or reviewers’ comments. We look forward to receiving your further evaluation of our manuscript.

Sincerely,

Florent de Vathaire, PhD

Head of the Radiation Epidemiology Group

Unit 1018 INSERM – CESP

Institut Gustave Roussy

39, rue Camille Desmoulins 

94805 Villejuif, France

Tel: +33 1 42 11 54 57

Fax: +33 1 42 11 53 15

Email: florent.devathaire@gustaveroussy.fr

Web: https://www.gustaveroussy.fr/ - https://cesp.inserm.fr/fr

All page and line numbering below refers to the re-submitted clean copy, word document, which has been line numbered, before re-submission.

Journal Requirements:

Our response: We thank the editor for your review. We have reviewed our reference list and have changed those that were found to be incorrect in order to fully meet PLOS ONE's style requirements.

Specific changes were made in the references:

Lines 401-403: “3. Freyer DR. Transition of Care for Young Adult Survivors of Childhood and Adolescent Cancer: Rationale and Approaches. J Clin Oncol. 2010;28: 4810–4818. doi:10.1200/JCO.2009.23.4278.”

Lines 419-422: “9. Berger C, Casagranda L, Faure-Conter C, Freycon C, Isfan F, Robles A, et al. Long-Term Follow-up Consultation After Childhood Cancer in the Rhône-Alpes Region of France: Feedback From Adult Survivors and Their General Practitioners. J Adolesc Young Adult Oncol. 2017;6: 524–534. doi:10.1089/jayao.2017.0019.”

Lines 471-473: “23. Nipp RD, Kirchhoff AC, Fair D, Rabin J, Hyland KA, Kuhlthau K, et al. Financial Burden in Survivors of Childhood Cancer: A Report From the Childhood Cancer Survivor Study. J Clin Oncol. 2017;35: 3474–3481. doi:10.1200/JCO.2016.71.7066.”

Lines 509-512: “34. van Dorp W, Haupt R, Anderson RA, Mulder RL, van den Heuvel-Eibrink MM, van Dulmen-den Broeder E, et al. Reproductive Function and Outcomes in Female Survivors of Childhood, Adolescent, and Young Adult Cancer: A Review. J Clin Oncol. 2018;36: 2169–2180. doi:10.1200/JCO.2017.76.3441.”

Lines 517-519: “36. Hjern A, Lindblad F, Boman KK. Disability in Adult Survivors of Childhood Cancer: A Swedish National Cohort Study. J Clin Oncol. 2007;25: 5262–5266. doi:10.1200/JCO.2007.12.3802.”

Additional Editor Comments:

Thank you for submitting your revised manuscript. Please consider the following additional comments in an additional revision and resubmission.

1. The wording of the revised title is awkward. Consider, "Healthcare expenditures among long-term survivors of pediatric solid tumors: Results from...."

Our response: Taking up your suggestion, we have changed the title of the article to: “Healthcare expenditures among long-term survivors of pediatric solid tumors: Results from the French Childhood Cancer Survivor Study (FCCSS) and the French network of cancer registries (FRANCIM).

2. The response to Comment 5 of Reviewer 1 should include the addition of this strength of the FCCSS study (long follow-up) to the Discussion.

Comment 5 – Reviewer 1. It took me some time to consider the inclusion of the diagnoses in the early years (they start in 1945) given the observation window for the outcome only starts in 2011, but I think this approach is strong as it produces estimates across the age spectrum.

Our response: We have added more details regarding Comment 5 of Reviewer 1.

Lines 359-360: “Another strength is the inclusion period starting in 1945, which allows us to study variations in cost across the age spectrum.”

3. The response to Comment 5 of Reviewer 2 is not clear and needs further explanation, including what is meant by the neuroblastoma group being "medically homogenous."

Our response: We clarified this as follows:

Lines 190-192: “Neuroblastoma was chosen as the referent for type of primary cancer variable since was one of the larger group of cancer of same histology”.

4. The term "paramedic" referring to costs needs clarification/definition.

Our response: We thank you for pointing out that “paramedic” were not clear enough. We have changed the names of this category to “Other health professionals visits” which is a more accurate translation:

We have also added a footnote to better define what’s included in the Table 2:

† Other medical professional visits included expenditures related to visits to podiatrist, optometrists, speech therapist, and others.

---

## [Editor Report · Decision Letter 2]

7 Apr 2022

Health care expenditures among long-term survivors of pediatric solid tumors: Results from the French Childhood Cancer Survivor Study (FCCSS) and the French network of cancer registries (FRANCIM)

PONE-D-21-17399R2

Dear Dr. de Vathaire,

We’re pleased to inform you that your manuscript has been judged scientifically suitable for publication and will be formally accepted for publication once it meets all outstanding technical requirements.

Kind regards,

David R Freyer

Guest Editor

PLOS ONE

Additional Editor Comments (optional):

Thank you for responding to the second round of comments, which have been satisfactorily addressed.
---

## [Editor Report · Acceptance letter]

16 May 2022

PONE-D-21-17399R2 

Health care expenditures among long-term survivors of pediatric solid tumors: Results from the French Childhood Cancer Survivor Study (FCCSS) and the French network of cancer registries (FRANCIM) 

Dear Dr. de Vathaire:

I'm pleased to inform you that your manuscript has been deemed suitable for publication in PLOS ONE. Congratulations! Your manuscript is now with our production department. 

Kind regards, 

on behalf of

Dr. David R Freyer 

Guest Editor

PLOS ONE